# Agroforestry Ecosystem Structure and the Stability Improvement Strategy in Control of Karst Desertification

Shilian Jiang [1,2], Kangning Xiong [1,2,*], Jie Xiao [1,2], Yiling Yang [1,2], Yunting Huang [1,2] and Zhigao Wu [1,2]

1   School of Karst Science, Guizhou Normal University, Guiyang 550001, China; sljiang@gznu.edu.cn (S.J.);
    mhxj47@gznu.edu.cn (J.X.); yyl980730@163.com (Y.Y.); yunting11183024@163.com (Y.H.);
    wuzg1224@163.com (Z.W.)
2   State Engineering Technology Institute for Karst Desertification Control, Guiyang 550001, China
*   Correspondence: xiongkn@gznu.edu.cn

**Abstract:** Agroforestry systems (AFS) are priority semi-natural ecosystems in fragile ecological zones. The complexity and diversity of their species structure play a crucial role in maintaining AFS stability. To explore the optimization of improvement strategies in AFS' structure and stability for control of karst desertification (KD), in this study, we chose typical desertification control areas in the southern China karst region. The study included homegarden (HG), agrisilviculture (ASV), and multipurpose woodlots (MWLs) as three AFS. We quantified the AFS' structural characteristics using descriptive statistics and spatial structure parameters. We used the fuzzy integrated evaluation method with structural and functional indicators as guidelines, and stand structure, plant species diversity, soil fertility, and environmental factors as first-level evaluation indicators. The entropy weight method calculates the weights of indicators at all levels. The fuzzy comprehensive evaluation method establishes an evaluation index system to evaluate the grading of AFS' stability. The results showed that: (i) The species composition of the AFS in the KD control areas had a simple structure, the overall diversity level was low, and the diversity level of herbaceous plants was better than that of woody plants. (ii) The overall distribution curves of diameter at breast height (DBH), tree height (TH), and crown width (CW) of woody plants in the AFS in the KD control areas were slight to the left, with a single-peaked pattern, mostly randomly and unevenly distributed in space, with a low degree of tree species isolation and relatively weak stand stability. (iii) There was variability in the stability classes of different types of AFS, overall reflecting the ranking HG > ASV > MWLs. (iv) When structural optimization was applied, corresponding measures can be taken according to farmers' wishes for different types of AFS and their primary business purposes. The improvement of stability depends mainly on the utility of the structural optimization applied coupled with positive human interference (for example, pruning, dwarfing, and dense planting). This study provides a scientific reference for maintaining the stability of AFS and promoting service provision.

**Keywords:** agroforestry; structure; species diversity; evaluation indicator systems; fuzzy integrated evaluation

## 1. Introduction

We are facing the threat of a rapid decline in global terrestrial biodiversity [1]. In contrast, habitat loss and environmental degradation due to population growth, agricultural intensification, and deforestation are the leading causes of loss of biodiversity (BD) and associated ecosystem functions and services [2]. Numerous studies have found that agroforestry is essential in maintaining biodiversity, improving the ecological environment in ecologically fragile areas, slowing down agricultural intensification, optimizing ecosystem services, and keeping farmers' livelihoods [3–8]. Compared to traditional intensive agroecosystems, agroforestry systems (AFS) have structural characteristics such as multi-level, multi-plant species composition and configuration in time and space, and the complexity of

AFS structure will influence the internal functions of the system and determine the supply of ecosystem services [9–11]. Regarding structure, higher connectivity, modularity, and nestedness can contribute to the dynamic stability or structural stability of intercropping ecosystems [12]. The more specific and complex the structure of an ecosystem, the more stable the design of the ecosystem [13,14]. Therefore, identifying AFS structural characteristics and analyzing the stability status of AFS to propose effective improvement strategies that will optimize the functions of AFS and enhance their service provisioning capacity are the focus and difficulty of current research [15].

The AFS are priority semi-natural ecosystems in fragile ecological zones. The proper structure has a crucial role in vegetation restoration, water conservation, soil conservation, increasing biodiversity, maintaining the system's stability, and improving the environment [16,17]. Agroforestry systems structure can be understood as an ecosystem attribute based on the composition of species and their quantitative relationships in the species structure, stratification in the vertical form, and mosaicism in the horizontal structure [18,19]. The AFS stability is dynamic, not static. It is a comprehensive characteristic of the system's ecological equilibrium state, such as system movement efficiency, resistance, and interactions between biotic and abiotic elements [20]. It reflects the interdependence and interaction relationships such as system structure and function over a certain period [9,21], and stability is a decisive factor in the system structure and service function [22]. Therefore, quantifying the AFS' structure, evaluating its stability, exploring the factors influencing its stability, and proposing strategies for structural optimization and stability improvement will be beneficial in maximizing the benefits brought by optimally adjusted AFS [23,24].

Karst, as a component of the world's major ecologically vulnerable areas [25], has become a priority research topic in the United Nations Sustainable Development Goals for restoring degraded ecosystems [26]. The local vulnerability of the karst ecosystems, combined with irrational human activities, has led to karst desertification (KD) as a critical ecological degradation problem in karst areas worldwide [27–30]. This has left karst regions with low disaster resilience and environmental capacity, limiting land use and economic development [31,32]. To promote economic growth and ecological restoration, scholars subsequently proposed AFS as one of the environmental restoration programs to apply in KD areas [33,34]. This would gradually be developed into homegarden (HG), which is mainly self-sufficient and distributed around the houses of the farmers. This increases the economic income of farmers and maintains ecological benefits as does agrisilviculture (ASV). It also has multiple uses, such as timber production, and provides various products, such as fruit, fodder, food, and industrial raw materials, such as multipurpose woodlots (MWLs). After years, it has been shown that the development of AFS in KD areas not only maximizes its soil and water conservation benefits [12,13,35] but also has a positive effect on improving the productivity of KD areas and maintaining soil faunal diversity [36,37]. It highlights the promising development of AFS in KD areas. In particular, the ecosystem services provided by AFS are essential for improving the overall ecosystem quality of the region and developing environmentally friendly industries that are compatible with the resources and environment. Studies indicate that the AFS structure is the essential dominant factor in providing ecosystem services [10,14], as it directly influences ecological processes and indirectly affects the supply of ecosystem services [11]. However, current studies on the structure and stability of AFS in the late stage are scarce [15].

Therefore, first, we focused on quantifying the species composition and diversity in the stand structure characteristics of woody plants in the AFS in the KD control areas. Second, taking the structure and function of the AFS as the standard layer, taking the stand structure, plants species diversity, soil fertility, and topographic factors as the first-level evaluation index indicators of stability, we used the fuzzy comprehensive evaluation method to construct a stability evaluation index system to evaluate the stability of different types of AFS in the KD control areas. Finally, the relationship between structure and stability and the influencing factors were explored, and practical strategies for structure optimization and stability improvement are proposed. To this end, we assumed that: (i) The

richness and diversity of species composition in the three AFS show MWLs > HG > ASV, but in terms of stand structure, HG is more reasonable, followed by MWLs and ASV are the worst. An urgent need is to take appropriate disturbance measures for optimization and adjustment. (ii) Considering the intensity of anthropogenic disturbance, the results of the stability evaluation showed the best stability of MWLs, followed by HG, and poor stability of ASV. (iii) From the weight values of each level of indicators, we can obtain that plant species diversity and stand structure were the main drivers of stability, and soil fertility and topographic factors were less influential, and we need to focus on species composition and their adaptability in the stability improvement strategy. To verify the above hypotheses, we quantified the structural characteristics of AFS in the KD region of southern China, evaluated its stability, and classified it into different classes. We explored the influencing factors of structure and stability and proposed strategies for structural optimization and stability improvement. It provides a scientific reference for maintaining AFS stability in the KD control areas and promoting the improvement of their service provisioning capacity.

## 2. Materials and Methods

### 2.1. Study Area

As early as China's 9th National Development Five-Year Plan in year, agroforestry (AF) was attempted to be adopted to combat the KD [33,34]. It was found that the blind value-addition of a population in karst areas, especially in mountainous areas where the slopes are dominated by agriculture, and where not enough attention is paid to forestry and pastoralism, resulted in an imbalance in the ratio of agriculture to other land uses. Forestry and pastoralism eventually led to the degradation of arable land, pastureland, and forest land, soil erosion, and increased floods and droughts. Furthermore, the deterioration of the ecological environment will become an inevitable consequence [38]. In turn, it has prompted various experts and scholars to consider different AF models in areas with varying levels of KD. Later, with further development, different types of AFS were gradually formed, such as homegarden (HG), mainly for farmers' self-sufficiency, and agrisilviculture (ASV), that primarily increases the economy and ecological restoration. In addition, it is not easy to distinguish the type of AFS (which has both the common characteristics HG and ASV) that combines the classification criteria of AFS by [39,40] to name it multipurpose woodlots (MWLs). Accordingly, based on the management experience of AFS in the study area, the representativeness and typicality of the study area, we selected the following three KD areas on the Guizhou plateau, to represent the overall ecological environment of karst in southern China (Figure 1).

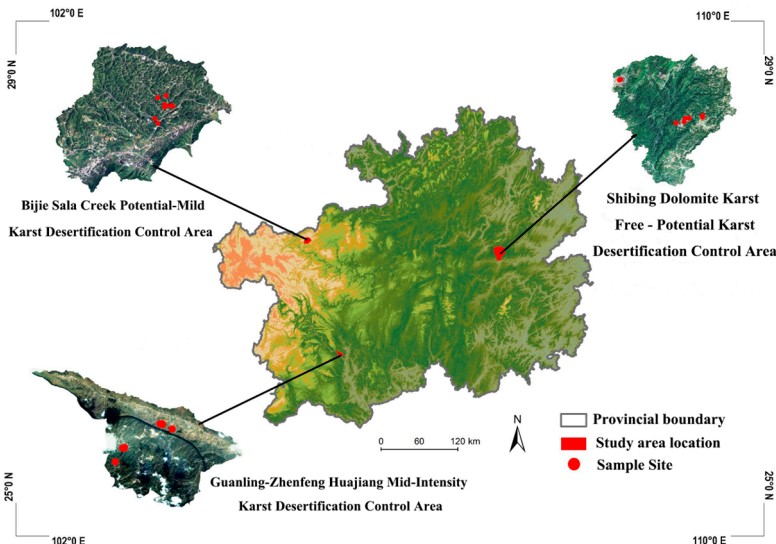

**Figure 1.** Study area.

First, the example area of potential–light KD in the karst plateau mountains of Bijie Salaxi was selected as the first study area (hereafter referred to as BJ). The study area (105°01′11″–105°08′38″ E, 27°11′09″–27°17′28″ N) has diverse geomorphic types and fragmented topography, and positive and negative topography such as peaks, depressions, funnels, dark rivers, and water caves are widely developed in the area. Elevations range from 1495 to 2200 m with a relative elevation difference of 705 m [41]. The northern subtropical humid monsoon climate has an average annual temperature of 12 °C and an average annual rainfall of about 984.4 mm. The area is mainly dryland, with a few paddy fields, dam terraces, and more slopes [42]. The AFS mainly comprises fruit trees such as walnuts, prickly pears, plums, apples, peaches, and loquats with crops such as ryegrass, white clover, maize, potatoes, and beans (Figure 2).

**Structural types of the AFS in the desertification control in Bijie Sala Creek**

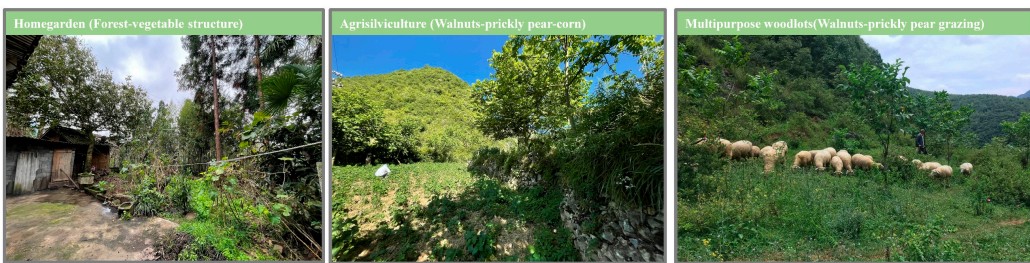

**Figure 2.** Schematic diagram of three types of AFS used in the control of KD in BJ.

Second, the Guanling-Zhenfeng Huajiang Karst Plateau Canyon moderate intensity KD area was selected as the second study area (hereafter referred to as HJ) (25°39′20″–25°41′20″ N, 106°37′30″–106°39′49″ E). There are various types of landforms and fragmented terrain. The favorable and hostile landscape, such as peaks, depressions, funnels, dark rivers, and water caves, are widely developed in the area. The elevation ranges from 370 to 1473 m with a relative elevation difference of 1103 m; the study area has a dry and hot southern subtropical valley climate, with warm and dry winters and springs and hot and humid summers and autumns, with an average annual temperature of about 18.4 °C and an average yearly precipitation of about 1100 mm [41]. The soil is predominantly developed on limestone with high soil fertility, but the soil layer is shallow and discontinuous, with poor water retention and drought tolerance [43]. The AFS in the KD area mainly comprises walnut, prickly pear, plum, apple, peach, loquat, and other fruit trees with ryegrass, white clover, and other forage grasses, corn, potatoes, beans, and other crops (Figure 3).

**Structural types of the AFS in the desertification control in Guanling-Chingfeng Huajiang**

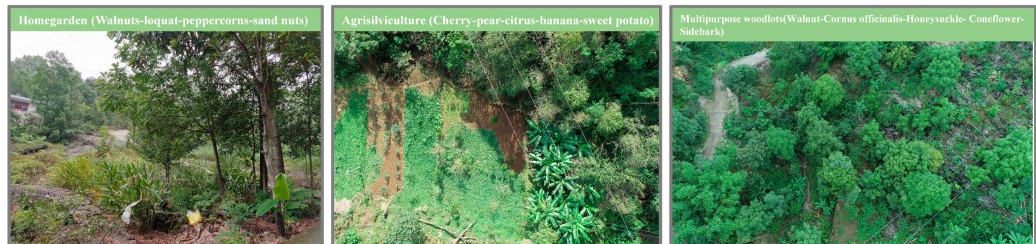

**Figure 3.** Schematic diagram of three types of AFS used in the control of KD in HJ.

Finally, the example area of no–potential KD in the Karst Plateau trough valley of Shibing was selected as the third study area (hereafter this is referred to as SB) (108°01′36″–108°10′52″ E, 27°13′56″–27°04′51″ N). The topography is high in the north and low in the south. Elevation ranges from 600 to 1250 m, with an average height of 912 m. It is a subtropical humid monsoon climate, with warm spring and cool summer, four seasons, an annual average temperature of 16 °C, and an average yearly rainfall of 1220 mm. The area has a thick soil layer, mainly lime soil, high soil fertility, and traditional agriculture

is well-developed. Therefore, the AFS in this KD area is composed chiefly of golden pear, peach, cherry, plum, and other fruit trees with medicinal plants or forage grasses such as yellow essence, prunus seeds, and white hyacinth (Figure 4).

**Structural types of the AFS in the desertification control in Shi Bing**

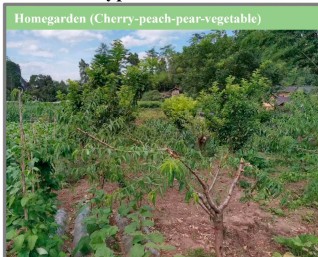 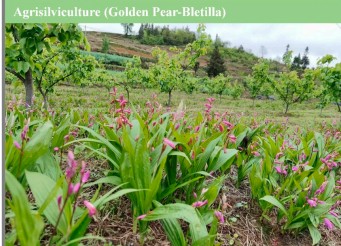 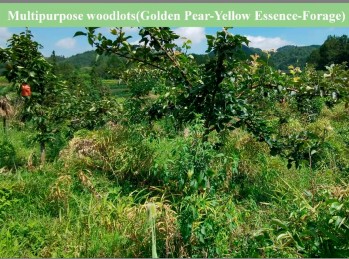

**Figure 4.** Schematic diagram of three types of AFS used in the control of KD.

*2.2. Sample Site Selection and Survey*

2.2.1. Sample Site Selection

Based on the above classification of different AFS for the KD areas, and through field survey by combining previous research results, we classified them into three types of AFS: HG, ASV, and MWLs. Three 20 m × 20 m test plots were set up for each AFS type in the three study areas, totaling nine plots per study area and 27 test plots for the three study areas.

2.2.2. Structural Investigation

First, each study plot's geographic coordinates and elevation were recorded with GPS (Garmin 639csx USA Kansas) [44] and the slope direction and gradient were measured with a slope meter (Supplementary Materials Table S1). Within each parcel, all tree species present in the sample square were recorded, the diameter at breast height of all plants was measured with a diameter at breast height ruler (cm), the crown width (m) was measured with a 50 m long measuring tape, and the tree height (m) was measured with a 15 m long telescopic height gauge. After that, two small diagonal sample squares with an area of 5 m × 5 m were selected for the survey, and all shrub species present in these sample squares were recorded, and their basal diameter (cm), crown width (m), and height (m) were measured. Finally, all herbaceous species occurring in five 1 m × 1 m small sample squares located at the corners and center of the sample square were measured and recorded. Their height (cm), the number of plants, multiplicity, and cover were also measured. For the above, species that were difficult to identify in the field, we harvested their leaves and brought them back to the laboratory to identify them concerning the references Flora of Yunnan, Flora of China, Higher Flora of China, Flora of China, and so on, to establish a database of information on plant species in each plot. We chose to conduct all the above surveys at the peak of plant growth (July 2022).

Therefore, combined with field investigation and research, we used Excel (Version 2019, Washington, DC, USA) to determine the plant species composition of the AFS in KD control areas and calculate the index parameters, SPSS (Version 25.0, Chicago, IL, USA) for descriptive statistical analysis, and Origin 2018(Version 9.0, Northampton, MA, USA) for mapping, aiming to explore the structural characteristics and stability of AFS.

2.2.3. Collection and Determination of Physical and Chemical Properties of Soils

In each 20 m × 20 m sample plot designed above, five small sample squares were distributed on the four corners and the center point. First, we collected samples of soil from 0–10 cm and 10–20 cm depth with a ring knife from the root system after removing the humus and brought it back to the laboratory to determine soil bulk density and capillary porosity. We collected 1 kg samples of soil separately from 0–10 cm and 10–20 cm depths in plastic bags and brought them back to the laboratory to determine the chemical properties

of soil organic matter, total nitrogen, total phosphorus, and total potassium. The details and equations for the collection and experimental operation of soil physical properties were calculated by referring to the method of Li Ke et al. [45]. Specific indicators are shown in Table 1.

**Table 1.** Indicators and measurement methods [45].

| Indicators | Measurement Methods |
| --- | --- |
| Soil pH (pH) | By water leaching-potentiometric method (National Standard GB7859-87) |
| Soil bulk density (SBD) | Soaking method and ring knife method |
| Soil capillary porosity (WFPS) | Soaking method and ring knife method |
| Soil organic matter (SOM) | National standard GB7857-87 oxidation of potassium dichromate -external heating method |
| Soil total nitrogen (TN) | National standard GB7173-87 semi-trace Kelvin method |
| Soil total phosphorus (TP) | Sodium hydroxide alkali fusion—molybdenum antimony anti-colorimetric method |
| Total potassium in soil (TK) | By flame photometry |

Soil Physical Indexes Were Determined

First, the fresh weight was determined by weighing the sample in the field, then it was brought back to the laboratory, and soaked for 12 h. Second, the ring knives and their contained samples were placed on absorbent paper for 2 h after full soaking and then weighed. Finally, the weighted ring knives were dried in an oven at 115 °C and then weighed once dry and the soil's physical properties were calculated [45]. The calculation formulae are:

$$\text{Soil bulk density (g·cm}^3\text{) (SBD)} = \text{Soil capacity (g·cm}^3\text{)} = \text{wet soil weight inside the ring knife/ring knife volume} \times (1 \pm \text{natural soil water content}) \tag{1}$$

$$\text{Soil capillary water holding capacity} = (\text{soil weight at 12 h of water absorption-dry soil weight})/\text{dry soil} \times 100 \tag{2}$$

$$\text{Soil capillary porosity (WFPS)} = \text{capillary water holding capacity} \times \text{capacity} \tag{3}$$

Soil Chemical Properties Determined

The soil brought back to the laboratory was air-dried and the chemical properties were determined by taking the determination method in Table 1, and calculated according to the following equation:

$$\text{Soil organic matter (g·kg}^{-1}\text{)} = \text{soil organic carbon (g·kg}^{-1}\text{)} \times 1.724 \tag{4}$$

$$\text{Soil organic carbon (SOC, g·kg}^{-1}\text{)} = \frac{\frac{c \times 5}{V_0} \times (V_0 - V) \times 10^{-3} \times 3.0 \times 1.1}{m \times k} \times 1000 \tag{5}$$

where: c—0.8000 mol·L$^{-1}$($K_2Cr_2O_7$), 5—the volume of potassium dichromate standard solution added (mL), $V_0$—the volume of $FeSO_4$ used for blank titration (mL), V—the volume of $FeSO_4$ used for sample titration (mL), 3.0—$\frac{1}{4}$ molar mass of carbon atoms (g-mol$^{-1}$), $10^{-3}$—Converting mL to L, 1.1—Oxidation correction factor, m—the mass of air-dried soil sample (g), k—the coefficient of converting air-dried soil to dried soil, 1.724—Average conversion factor for conversion of soil organic carbon to soil organic matter.

$$\text{Soil total nitrogen (TN, g·kg}^{-1}\text{)} = \frac{(V_0 - V) \times c\left(\frac{1}{2}H_2SO_4\right) \times 14.0 \times 10^{-3}}{m} \times 10^3 \tag{6}$$

where: V—the volume of the acid standard solution used in the titration of the test solution (mL); $V_0$—the volume of the acid standard used in titration blank (mL); c—0.01 mol·L$^{-1}$

($K_2Cr_2O_7$) or the concentration of HCL standard solution; 14.0—the molar mass of nitrogen atoms (g·mol$^{-1}$); $10^{-3}$—Converting mL to L, m—the mass of the dried soil sample (g).

$$\text{Soil total phosphorus}\left(\text{TP, g·kg}^{-1}\right) = \rho \times \frac{V_1}{m} \times \frac{V_2}{V_3} \times 10^{-3} \times \frac{100}{100-H} \qquad (7)$$

where: ρ—the mass concentration of phosphorus in the solution of the sample to be measured (mg·L$^{-1}$) as found from the calibration curve; m—weighing sample mass (g); $V_1$—the volume of the fixed volume of the sample after melting (mL); $V_2$—the volume of solution fixation when developing color (mL); $V_3$—volume dispensed from the molten sample after volume fixing (mL); $10^{-3}$—Conversion factor for converting mg·L$^{-1}$ concentration units to kg mass; $\frac{100}{100-H}$—Conversion factor for converting air-dried soil to dried soil; H—the percentage of moisture content in air-dried soil.

$$\text{Soil total potassium}\left(\text{TK, g·kg}^{-1}\right) = \frac{\rho \times MVC \times SM}{m \times 10^6} \times 1000 \qquad (8)$$

where: ρ—the mass concentration of K in the solution to be measured from the standard curve (μg·mL$^{-1}$); MVC—Constant volume of the measuring solution; SM—Score multiplier; m—weighing sample mass (g); $10^6$—Conversion of μg to g divisor.

### 2.2.4. Determination of the Rate of Rock Outcrop

We collected rock outcrops rate specimens by the mechanical pointing method and photography method (Figure 5). First, holding a 1 m long bamboo pole, we walked along the edge of the 20 × 20 m sample, clicked on the ground every meter, determined whether the point was in contact with a rock, and counted the rocks. The absence of rocks was not measured, rather it was the number of sample points clicked on rock outcrops that were counted [46]. The rock exposure rate was calculated as the rock outcrops rate (BRR) = the number of sample points in contact with rocks/total number of sample points.

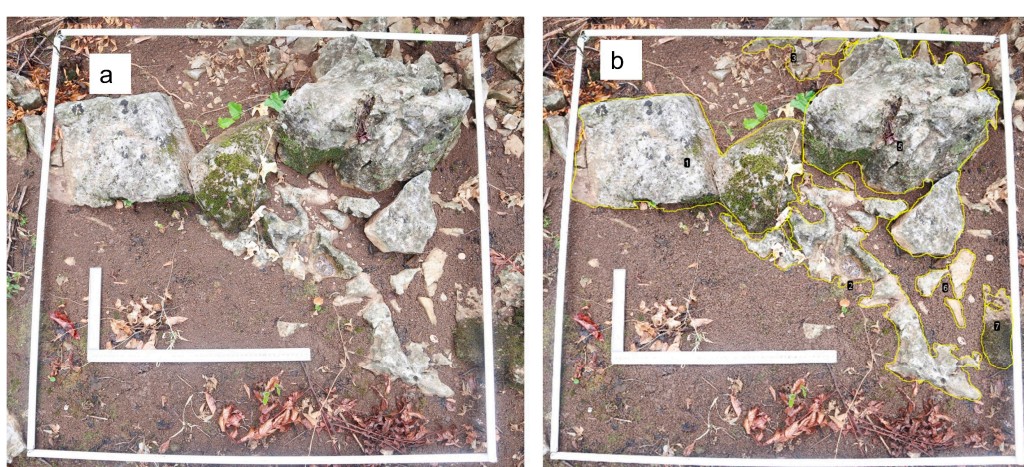

**Figure 5.** Measuring the percentage of surrounding limestone rock outcrops in AFS. The (**a**) is the original photo of 1 m × 1 m, and the (**b**) shows the rock area that was measured in the 1 m × 1 m.

Second, the size and area of rocks covered by the soil surface (1 × 1 m) were determined by digital photography and later by using ImageJ processing software (Version 1.8.0.112, New York, NY, USA) [47]. The above data were collected at the peak of plant growth (July 2022).

### 2.3. Parameter Calculation and Model Construction

#### 2.3.1. Calculation of Parameters of AFS Structure

This research used the spatial structure analysis method of a stand with one reference tree and four nearest neighboring trees. The angular scale (W), mixing degree (M), and

stand size ratio (U) were selected to establish the distribution of three spatial structure parameters to analyze the spatial distribution pattern [48,49], the degree of species isolation, and the degree of individual differentiation of woody plant distribution in ASF of the KD control area.

### Angular Scale

Angular scale ($W_i$). The angular scale reflects the distribution pattern of individual trees in an agroforest and refers to the proportion of $\alpha$ angles smaller than the standard angle $\alpha_0$ ($\alpha_0 = 72°$) to the four $\alpha$ angles examined [48], which is calculated by equation.

$$W_i = \frac{1}{n} \sum_{j=1}^{n} Z_{ij} \tag{9}$$

where: $W_i$ is the angular scale of reference tree I; the $z_{ij}$ is a discrete variable; $z_{ij} = 1$ when the jth $\alpha$-angle is smaller than the standard angle $\alpha_0$, and $z_{ij} = 0$ vice versa.

The five values and significance of $W_i$: $W_i = 0$, indicates that the stand is particularly uniformly distributed; $W_i = 0.25$, indicates that the stand is uniformly distributed; $W_i = 0.5$ indicates that the stand is randomly distributed; $W_i = 0.75$, indicates that the stand is unevenly distributed; and $W_i = 1$ indicates that the stand is very unevenly distributed. For the overall stand mean, the range of the random distribution is (0.475, 0.517), and for the overall stand mean ($\overline{w}$), the range of random distribution is within (0.475, 0.517), with $\overline{w} > 0.517$ being a clumped distribution and $\overline{w} < 0.475$ being a uniform distribution.

### Mixing Degree

Mixing degree ($M_i$). The degree of admixture was used to examine the spatial isolation of tree species. It is described as the proportion of the four nearest neighbors of reference tree I that is not of the same species as the proportion of individuals in the four nearest neighbors of reference tree i that are not of the same species as reference tree i. It was calculated by see equation:

$$M_i = \frac{1}{n} \sum_{j=1}^{n} V_{ij} \tag{10}$$

where: $M_i$ is the mixing degree of reference tree I; the $V_{ij}$ is the discrete variable $v_{ij} = 1$ when the reference tree i is not the same species as the jth neighboring tree, $v_{ij} = 0$, otherwise, $v_{ij} = 0$ [49].

The five values and meanings of $M_i$: $M_i = 0$, indicates that the stand is zero-mixed; $M_i = 0.25$, indicates that the stand is weakly mixed; $M_i = 0.5$, indicates moderate mixing; $M_i = 0.75$, indicates strong mixing; $M_i = 1$, indicates that the stand is very strong. For the overall stand mean value ($\overline{M}$), the larger the value indicates the higher the degree of species isolation, and the more stable the stand.

### Size Ratio

Size ratio ($U_i$). The size ratio refers to the ratio of the number of neighboring trees with diameters larger than the reference. The ratio of the number of neighboring trees with a diameter at breast height greater than that of the reference tree to the four nearest neighboring trees examined. The calculation method is shown in equation.

$$U_i = \frac{1}{n} \sum_{j=1}^{n} K_{ij} \tag{11}$$

where: $U_i$ is the size ratio of the reference tree I; the $k_{ij}$ is the discrete variable If the neighboring tree j is smaller than the reference tree i, $k_{ij} = 0$, otherwise, $k_{ij} = 1$ [48].

The five values and meanings of $U_i$. $U_i = 0$ indicates that the reference tree is dominant; $U_i = 0.25$, indicates that the reference tree is subdominant; $U_i = 0.75$, indicates that the reference tree is inferior; $U_i = 1$, indicates that the reference tree is inferior. It can be seen that the size ratio quantifies the relationship between the reference tree and its neighbors, and the lower the value ($U_i$), the fewer neighbors have a larger diameter at breast height

than the reference tree. For the mean size-ratio value ($\overline{U}$), the statistics by species can better understand the competition among species in the stand, and there are five values and meanings: when the $\overline{U} = 0$, the species is preferred. When $0 < \overline{U} \leq 0.33$, the species is suboptimal. When $0.33 < \overline{U} \leq 0.67$, the species is in the intermediate state. When $0.67 < \overline{U} < 1$, the species is in an inferior condition. When $\overline{U} = 1$, it means that the species is inferior. Among them, those in superior and suboptimal states are dominant species, and those in inferior and absolute inferior states are inferior tree species.

### 2.3.2. Species Diversity

Based on the field survey data, the plant species and animal species in the AFS were surveyed, and the importance value (IV) of plant species in the sample site was calculated, as well as its species diversity index, with the importance value IV = (relative abundance + relative frequency + relative dominance)/3 [44].

$$\text{Shannon Diversity Index } H' : \ H' = -\sum_{i=1}^{s} P_i ln p_i \tag{12}$$

$$\text{Pielou Uniformity Index } E : \ E = \frac{H'}{lnS} \tag{13}$$

$$\text{Simpson dominance index C} : \ C = \sum_{i=1}^{s} \frac{N_i(N_i - 1)}{N(N - 1)} \tag{14}$$

$$\text{Margalef Richness Index D} : \ D_{MG} = \frac{S - 1}{lnN} \tag{15}$$

where: in Equations (12)–(15): Pi is the frequency; $P_i = N_i/N$, S is the number of taxa; Ni is the number of individuals of the "i" taxon; and N is the total number of individuals of all taxa.

### 2.3.3. Establishment of Evaluation Index System
Selection Criteria and Principles of Evaluation Indicators

As a semi-natural ecosystem, the AFS cannot escape the interaction between material cycles and energy transfer with the surrounding environmental conditions. It makes them into a non-linear zone far from equilibrium. To better understand the structure and stability of AFS, the concept of multi-indicator evaluation based on information entropy theory can provide us with a measure of uncertainty or confusion of information in AFS so that we can better rely on important information from them to achieve the improvement of AFS stability. Therefore, we comprehensively consider the actual situation of AFS in KD areas. To ensure the system reliability of the evaluation system, we aimed to evaluate the AFS stability in KD control areas and develop the principles and criteria for constructing the index system. (Supplementary Materials Tables S2 and S3).

Establishment of Evaluation Indicators

We divided the evaluation factors into the target, criterion, and indicator layers according to the dissipative structure information entropy theory. The target layer is agroforestry ecosystem stability. The criterion layer is the AFS structure and KD standing environment factors. The indicator layer is divided into primary indicators and secondary indicators. The first-level hands are four, including forest stand structure, species diversity, soil fertility, and topographic factors. There are 20 secondary indicators, including angular scale, hybridization, size ratio, depression, tree height, crown width, plant richness index, diversity index, uniformity index, dominance index, soil organic matter, total soil nitrogen, total soil phosphorus, total soil potassium, soil bulk, soil capillary porosity, soil pH, slope, elevation, and rock outcrops rate. Based on the above target layer, criterion layer, and indicator layer, we constructed the evaluation index system (Table S4 in Supplementary Materials), which provides support for further construction of the weight matrix and affiliation matrix, and fuzzy matrix based on the indicators in the criterion layer and target layer, respectively.

Among them, the primary index is expressed as U= {$U_1$, $U_2$, $U_3$..., $U_n$}, and the secondary index is expressed as $U_i$= {$U_{i1}$, $U_{i2}$, $U_{i3}$, ..., $U_{in}$} (i = 1, 2, 3, ..., n), and Uin denotes the nth index of the ith level of evaluation [50,51].

Calculation of Indicator Weight Values

a. Assume that there are mth agroforestry samples and nth evaluation indicators.

From this, the original matrix X′ = [X′$_{ij}$]$_{mn}$ for stability evaluation can be established as follows:

$$X' = \begin{bmatrix} x'_{11} & x'_{12} & \cdots & x'_{1n} \\ x'_{21} & x'_{22} & \cdots & x'_{1n} \\ \vdots & \vdots & \vdots & \vdots \\ x'_{m1} & x'_{m2} & \cdots & x'_{mn} \end{bmatrix} \tag{16}$$

where: $x'_{ij}$ is the value of the j evaluation indicator in the ith AFS sample.

b. Data standardization.

The data of the sample matrix X′ is dimensionless, and this paper uses the extreme value method to process:

When the indicator data response is a positive effect:

$$X_{ij} = \frac{x'_{ij} - min(x'_{ij})}{max(x'_{ij}) - min(x'_{ij})} \tag{17}$$

When the indicator data response is a negative effect:

$$X_{ij} = \frac{max(x'_{ij}) - x'_{ij}}{max(x'_{ij}) - min(x'_{ij})} \tag{18}$$

where: $X_{ij}$ is the normalized value, where $1 \leq i \leq m$, and i is an integer; max($X'_{ij}$) and min($X'_{ij}$) are the maximum and minimum values in row i of the matrix X′, respectively.

The matrix X after data normalization is:

$$X = \begin{bmatrix} x_{11} & x_{12} & \cdots & x_{1n} \\ x_{21} & x_{22} & \cdots & x_{2n} \\ \vdots & \vdots & \vdots & \vdots \\ x_{m1} & x_{m2} & \cdots & x_{mn} \end{bmatrix} \tag{19}$$

c. Calculate the entropy value of the jth evaluation index.

The formula for calculating the entropy value of each evaluation index is:

$$E_j = -\frac{\sum_{i=1}^{m} f_{ij} ln f_{ij}}{lnm} \tag{20}$$

$$f_{ij} = \frac{x_{ij}}{\sum_{i=1}^{n} x_{ij}} \tag{21}$$

where: $E_j$ is the entropy value, $E_j \geq 0$; $f_{ij}$ is the frequency of evaluation index j in the ith sample, $0 \leq f_{ij} \leq 1$.

d. Calculation of weight values:

$$W_j = \frac{1 - E_i}{\sum_{i=1}^{n}(1 - E_i)} - \frac{1 - E_i}{n - \sum_{i=0}^{n} E_i} \tag{22}$$

where: $0 \leq W_j \leq 1$, $\sum_{i=1}^{n} W_i = 1$. After calculating the weight of each index, the entropy weight matrix $W_i$= {$W_{i1}$, $W_{i2}$, ..., $W_{i3}$}, for the second level evaluation index is determined first, and then the entropy weight matrix W= {$W_1$, $W_2$, ..., $W_n$}, for the first level evaluation index is determined.

Establishment of the Evaluation Set

Based on the available research results and the nature and specific conditions of the evaluated objects, the evaluation set V = {V$_1$, V$_2$, . . . , V$_k$} is created for the first and second levels of evaluation indicators, where k is the number of evaluation levels and k = 5. The evaluation indexes were classified into I, II, III, IV, and V classes, which were excellent, good, moderate, poor, and very poor, respectively. There is no uniform classification standard for some evaluation indicators, so this paper uses the extreme difference method to classify them according to the actual measurement results (Supplementary Materials Table S5. Evaluation Table S6).

Creation of Fuzzy Matrix

According to the established evaluation set, V distinguishes the affiliation degree r$_{ij}$($0 \leq r_{ij} \leq 1$)) of the set to which each evaluation factor Ui belongs. The descending semi-trapezoidal affiliation function is used to calculate the affiliation of negative effect indicators; the ascending semi-trapezoidal affiliation function is used to calculate the association of positive effect indicators. The fuzzy relationship matrix R (i.e., the affiliation matrix) is obtained according to the established affiliation function.

$$R = \begin{bmatrix} r_{11} & r_{12} & \cdots & r_{1k} \\ r_{21} & r_{22} & \cdots & r_{2k} \\ \vdots & \vdots & \vdots & \vdots \\ r_{n1} & r_{n2} & \cdots & r_{nk} \end{bmatrix} \tag{23}$$

where: r$_{ij}$ is the affiliation degree of the "j" evaluation level to which the "i" evaluation index belongs.

Comprehensive Evaluation

First, the fuzzy comprehensive evaluation of the second-level indicators; the fuzzy evaluation vector S$_i$ of the second-level indicators are obtained.

$$S_i = w_i \cdot R_i = \{w_{i1}, w_{i2}, \ldots, w_{ij}\} \cdot \begin{bmatrix} r_{11} & r_{12} & \cdots & r_{1k} \\ r_{21} & r_{22} & \cdots & r_{2k} \\ \vdots & \vdots & \vdots & \vdots \\ r_{n1} & r_{n2} & \cdots & r_{nk} \end{bmatrix} \tag{24}$$

where: $S_i$ is the affiliation vector of the evaluation set V to which the second-level evaluation factor Ui belongs, $1 \leq i \leq 3$, and i is an integer. Then the fuzzy comprehensive evaluation is calculated for the first-level indexes.

$$A = W \cdot S \tag{25}$$

where: W is the weight vector of the first-level indicators; S is the affiliation vector of the evaluation set *V* to which the first-level evaluation factor *U* belongs; A is the affiliation vector of the evaluation set *V* to which the study area's AFS stability belongs. According to the principle of maximum affiliation, the affiliation level of A can be determined.

## 3. Results

### 3.1. Plant Species Components

Components are the elements that make up the system. The most important feature of the AFS, unlike agricultural (mainly plantation) and forestry systems, is that they have many structural components; as the number of components increases, the diversity, complexity, and stability of the system increase, and the processes of material cycling and energy flow in the system change, and productivity increases. In the case of AFS, the main components are crops and forest trees, whose presence and quantity play an essential role

in maintaining the balance and stability of the ecosystem. Therefore, this section aimed to explore the species composition and diversity of AFS in KD control areas.

3.1.1. Types of AFS Structure

The fieldwork and statistics revealed that the primary structure types of the homegarden (HG), the agrisilviculture (ASV), and the multipurpose woodlots (MWLs) in the three study areas had significant differences.

Firstly, in terms of the HG, the BJ study area vegetation was mainly comprised of walnut (*Juglans regia* L.), prickly pear (*Rosa roxburghii* Tratt.), apple (*Malus pumila* Mill.), plum (*Prunus salicina* Lindl.), loquat (*Eriobotrya japonica* Lindl), and other woody plants and vegetables with understory beekeeping, understory chicken farming, and different primary structure types such as walnut-prickly pear-beekeeping, walnut-apple-chicken keeping, apple-plum-vegetable, farming etc. The spatial relationship is mainly clumped mixed species and contour belt-type hybrid species. The HJ study area vegetation primarily comprised walnut, loquat, pear (*Pyrus* spp.), pepper (*Zanthoxylum bungeanum* Maxim.), persimmon (*Diospyros kaki* Thunb.), cherry (*Prunus pseudocerasus* Lindl.), citrus (rosa peach clementine), and other woody plants combined with amomi fructus (*Amomum villosum* Lour.), sweet potato (*Ipomoea batatas* (L.) Poir.), cabbage (*Brassica rapa* var. Glabra regel), eggplant (*Solanum melongena* L.), and other crops such as walnut-loquat-pear-sand-potato, walnut-pepper-sand, citrus-persimmon-cherry-vegetable, etc. The spatial relationships were mainly vertical spatial structures and clumped mixed species. In SB, which is primarily composed of woody plants such as peach (Pesca), cherry, and plum in combination with crops such as cabbage, eggplant, pepper, and tomato (*Solanum tuberosum* L.). Its primary structure types peach-plum-cherry-vegetable, peach-corn-vegetable, and peach-potato-vegetable, were mainly dominated by the spatial relationship of clumped mixed species.

Secondly, as far as the ASV is concerned, the BJ study area mainly comprised woody plants such as walnut, apple(malus), and prickly pear with ryegrass (*Lolium perenne* L.), maize (*Zea mays* L.), white clover (*Trifolium repens* L.), etc. The primary structure types of walnut-apple-white clover, walnut-prickly pear-corn, walnut-corn-rye grass, etc., are mainly dominated by clumped mixed species with vertical spatial structure. The HJ study area vegetation mainly consists of pepper, dragon fruit (*Hylocereus undulatus* Britt), plum, citrus, and other woody plants with corn, sugarcane (*Saccharum officinarum* L.), peanut (*Arachis hypogaea* L.), and different structural types such as pepper-dragon fruit-sugarcane, pepper-plum, pepper-dragon fruit, citrus-plum-corn, intercropping and double-layer structure. The SB study area vegetation mainly consists of golden pear (*Pyrus pyrifolia* (Burm.f.) Nakai), peach, and other woody plants with pasture (*Arthraxon* P.Beauv.), corn, roasted tobacco (*Nicotiana tabacum* L.), and different structural types of golden pear-pasture, peach-roasted tobacco peach-corn, mainly for the double-layer structure.

Finally, regarding the MWLs, the BJ study area vegetation is mainly comprised of walnut, prickly pear, firethorn (*Pyracantha fortuneana* (Maxim.) H.L.Li), single flower (*Hypericum monogynum* L.), lacquer tree (*Rhus verniciflua* Stokes), and other woody plants combined with pasture and different structural types such as walnut-prickly pear-pasture, prickly pear-goldenrod-pasture, walnut-firethorn-lacquer tree-pasture, which are mainly clumped mixed species. The HJ study area vegetation primarily consists of woody plants such as walnuts, paper mulberry (*Broussonetia papyrifera* (L.) L'Hér. ex Vent.), medicinal evodia fruit (*Euodia* sp.), oriental arborvitae (*Platycladus orientalis* (L.) Franco), loquat (*Eriobotrya japonica* (Thunb.) Lindl.) with honeysuckle flower (*Lonicera* L.), banana (*Musa* × *paradisiaca* L.), bamboo (Bambuseae), etc. The structure types of walnuts-honeysuckle flower-medcinal evodia fruit, walnuts-paper mulberry-banana, oriental arborvitae-walnuts-honeysuckle flower, etc., are mainly in the form of clumped mixed planting and intercropping structure. The study area of SB mainly consists of golden pear combined with perennial herbs such as many flower Solomonseal rhizome (*Polygonatum sibiricum* Redouté), hyacinth orchid (*Bletilla striata* (Thunb.) Rchb.f.), heterophylly falsestarwort root (Radix Pseudostellariae), grass family (Gramineae). Such as the golden pear- many floSolomonsealseal rhizome-grass

families, golden pear- hyacinth orchid-grass family, golden pear- heterophylly falsestar-wort root-grass family, and other structural types, mainly intercropping mixed and double layer structure.

3.1.2. Species Composition and Importance Values

In the AFS, crops and forest trees are notable species that play a key role in the balance and stability of the ecosystem. First, forest trees are an integral part of AFS, providing wood, fiber, and other biomass needed by humans and critical ecological services to the ecosystem. Second, crops offer food and other necessities humans need while providing ecological benefits, promoting ecosystem diversity, and providing food and habitat for other organisms. Developing AFS can also contribute to ecosystem diversity by providing food and habitat for other organisms. Statistical analysis of the species composition of the three study areas resulted in the following (Supplementary Materials Tables S7 and S8).

Firstly, in the HG, the BJ study area had 29 species of plants belonging to 20 families and 29 genera, among which nine species of plants of Asteraceae accounted for 21.95% of the total, four species of plants of Gramineae accounted for 9.76% of the total, two species of plants of Labiatae, Caryophyllaceae, Polygonaceae, and Urticaceae respectively, and 4.88% of the total respectively, and the rest of the families had only one species of plants. The HJ study area had 19 species of plants belonging to 15 families and 17 genera, among which two *Asteraceae* and *Leguminosae*, respectively, accounted for 10.53% of the total, and the rest had only one species. The SB study area had a total of 43 species of plants belonging to 19 families and 42 genera, including seven species of Asteraceae, accounting for 16.28% of the total, four species of Labiatae and Solanaceae, accounting for 9.30% of the total, three species of Gramineae, accounting for 6.98%, Rosaceae, Primulaceae, liliaceae, and Polygonaceae all had two species, accounting for 4.65% of the total. In comparison, the rest of the families had only one species. Taken together, the above species structures of Asteraceae, Gramineae, and Labiatae were present in both the BJ and SB study areas, indicating their greater dominance in the development of HG, while the HJ study area showed more significant differences.

Secondly, in the ASV, the BJ study area had 27 species of plants belonging to 19 families and 27 genera, including five species of Asteraceae (16.22% of the total), four species of Gramineae (13.51% of the total), three species of Rosaceae (10.81%) and three species of Labiatae (8.11%). The SB study area had 21 species of plants, belonging to 10 families and 21 genera, including four species of grasses, accounting for 19.05% of the total, four species of Labiatae, and four species of Asteraceae, accounting for 9.52% of the total, and only one species of plants in each family. In the above AS species structure, plants of Asteraceae were all present, indicating that they had a more significant advantage in developing AS.

Finally, in the MWLs, the BJ study area had 30 plant species belonging to 13 families and 21 genera, among which seven species of Asteraceae accounted for 26.67% of the total, four species of Gramineae accounted for 13.33% of the total, two species of Leguminosae accounted for 6.67% of the total, and the rest had only one species of plants in each family. The HJ study area had 14 plant species, belonging to 10 families and 12 genera, among which two species of Asteraceae and two species of Gramineae accounted for 14.29% of the total, and the rest had only one species of plant in each family. The SB study area had 32 species of plants belonging to 17 families and 22 genera, including five species of Asteraceae, accounting for 15.63% of the total, four species of Gramineae, accounting for 12.50% of the total, two species of Amaranthaceae, accounting for 9.38% of the total, and two species of Xanthate, accounting for 6.25% of the total. The rest of the families had only one species of plants. In the above species structure, Asteraceae were present in all three study areas, indicating that Asteraceae have a tremendous advantage in developing AFS.

3.1.3. Plant Diversity

The analysis of plant species diversity (Supplementary Materials Table S9) reveals the plant species diversity for the different types of AFS in the KD areas. Overall, the AFS

formed by KD has a simpler structure and lower species diversity and generally shows a higher level of herbaceous plant species diversity than woody plant species diversity. In particular, it was found that the AFS formed by the KD mainly consisted of a two-layer structure consisting of trees and herbs or shrubs and herbs, indicating that their hierarchical structure was relatively like HG. The H' diversity index of different types of the AFS in the various study areas was ranked BJ > HJ > SB, indicating that the overall species diversity level in the BJ study area was higher than that in the other two study areas. The evenness index E reflected the uniformity of plant species in AFS, and the overall performance was ranked HJ > BJ > SB, indicating that the general distribution of AFS in the HJ study area was more uniform. The C dominance index showed an opposite pattern to the H diversity index. The SB study area with a lower species diversity index value had a lower overall diversity level and higher dominance.

### 3.2. Forest Stand Structure in the AFS

For the AFS, the main components are crops and perennial woody plants. It is an artificial system configured with perennial woody plants and produced in a particular time and space to imitate the natural ecosystem according to individual functional needs. It has not only the properties of the natural ecosystem but also the properties of the plantation ecosystem [9,16]. It has also been pointed out that in the study of AFS, not only the horizontal properties (such as species composition and their diversity) should be considered, but it is also necessary to pay attention to their vertical spatial structural characteristics (such as the spatial location of woody plants and their relationships) to make possible to maintain the AFS total development. On the one hand, the diameter at breast height, crown width, and tree height in the non-spatial structure are essential indicators of woody plant growth and ecosystem health. They can provide information about the stand structure and species diversity of AFS, which can better reveal the structural characteristics of AFS and are essential for the adequate protection and maintenance of healthy development of ecosystems. On the other hand, spatial structural characteristics describe the properties of the spatial location of forest trees and their arrangement relationships, which largely determine the functions played by agroforestry. A complete understanding of its spatial structural characteristics is necessary to develop more appropriate agroforestry management strategies to guide its management practices.

Therefore, we analyzed the characteristics of the AFS structure in the KD control areas from both non-spatial and spatial systems of forest stands to provide theoretical support for optimizing the AFS structure.

### 3.2.1. Nonspatial Structure

In this section, we describe the nonspatial structural characteristics of three agroforestry systems (AFS) types. They are homegarden (HG), agrisilviculture (ASV), and multipurpose woodlots (MWLs).

#### Nonspatial Structural Characteristics of the HG

In the HG (Figure 6), the values for diameter at breast height, tree height, and crown width of agroforestry woody plants in the BJ study area were $8.05 \pm 1.23$ cm, $4.21 \pm 0.61$ m, and $2.94 \pm 0.31$ m, respectively. The values for diameter at breast height, tree height, and crown width of the HJ study area woody plants were $8.30 \pm 1.23$ cm, $4.16 \pm 0.47$ m, and $3.14 \pm 0.34$ m, respectively. In the SB study area, the values for woody plants' diameter at breast height, tree height, and crown width were $6.01 \pm 0.86$ cm, $3.40 \pm 0.38$ m, and $2.89 \pm 0.22$ m, respectively. Among them, the values for diameter at breast height, tree height, and crown width of woody plants in the BJ and HJ study areas were larger than those in the SB study area, indicating that woody plants in the BJ and HJ study areas were more luxuriant. In addition, from the perspective of kurtosis and skewness, the overall distribution curve of the non-spatial structure of woody plants in the BJ and HJ study areas was leftward and spiky. In contrast, the canopy width distribution of woody plants in the

SB study area was more dispersed, and the peaks were flat. The woody plants in the BJ and HJ study areas showed more concentrated distribution characteristics, while those in the SB study area were more dispersed.

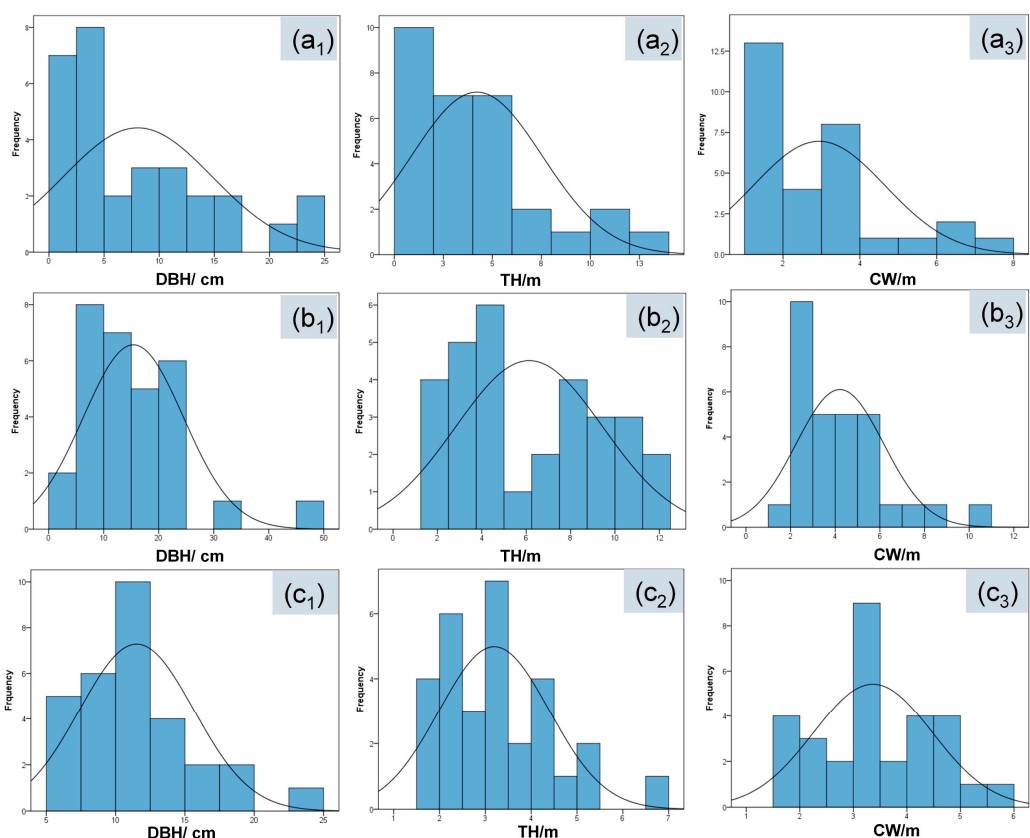

**Figure 6.** Nonspatial structure in HG. DBH, TH, and CW in the figure indicate the diameter at breast height (cm), tree height (m), and crown width (m), respectively (the same in the figure below); in addition, (**a₁**–**a₃**) indicate the BJ, (**b₁**–**b₃**) indicate the HJ, (**c₁**–**c₃**) indicate the SB.

Nonspatial Structural Characteristics of the ASV

In the ASV (Figure 7), the values for diameter at breast height, tree height, and crown width of agroforestry woody plants in the BJ study area were $15.39 \pm 1.66$ cm, $6.14 \pm 0.61$ m, and $4.21 \pm 0.36$ m, respectively. The values for diameter at breast height, tree height, and crown width of agroforestry woody plants in the HJ study area were $8.59 \pm 1.02$ cm, $3.10 \pm 0.52$ m, and $2.64 \pm 0.28$ m, respectively. The values for diameter at breast height, tree height, and crown width of agroforestry woody plants in the SB study area were $10.93 \pm 0.88$ cm, $4.13 \pm 0.49$ m, and $2.99 \pm 0.27$ m, respectively. The diameter values at breast height, height, and crown width of woody plants in the SB study area were $10.93 \pm 0.88$ cm, $4.13 \pm 0.49$ m, and $2.99 \pm 0.27$ m. In conclusion, among the ASV, the parameters diameter at breast height, tree height, and crown width of woody plants in the BJ study area were the largest, while those for the diameter at breast height, tree height, and crown width of woody plants in the HJ study area were the smallest. The SB study area was at an intermediate level. In addition, the kurtosis and skewness values of diameter at breast height, tree height, and crown width in the three study areas were greater than zero except for the HJ tree height distribution, which was less than zero, indicating that the overall distribution curve of the non-spatial structure of this type of ASV was slightly left and spiky.

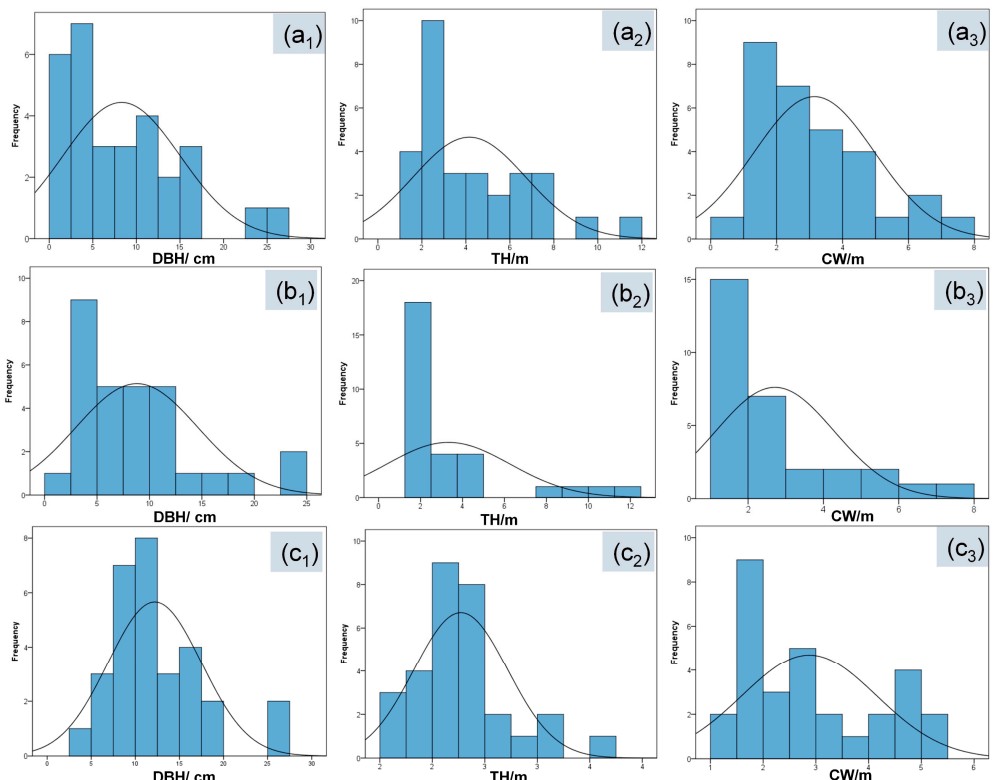

**Figure 7.** Nonspatial structure in ASV. DBH, TH, and CW in the figure indicate the diameter at breast height (cm), tree height (m), and crown width (m), respectively (the same in the figure below); in addition, ($a_1$–$a_3$) indicate the BJ, ($b_1$–$b_3$) indicate the HJ, ($c_1$–$c_3$) indicate the SB.

Nonspatial Structural Characteristics of the MWLs

In the MWLs (Figure 8), the values for diameter at breast height, height, and crown width of agroforestry woody plants in the BJ study area were 11.52 ± 0.75 cm, 3.19 ± 0.22 m, and 3.37 ± 0.20 m, respectively. The values for diameter at breast height, tree height, and crown width of the HJ study area woody plants were 11.34 ± 1.25 cm, 2.68 ± 0.42 m, and 2.97 ± 0.38 m, respectively. The values for diameter at breast height, tree height, and crown width of the SB study area woody plants were 9.30 ± 0.81 cm, 2.04 ± 0.11 m, and 1.97 ± 0.11 m, respectively. Among them, the kurtosis values of the canopy width distribution of the diameter at breast height of the BJ study area woody plants were less than zero, and the skewness values were more significant than zero, which indicated that the canopy width distribution was more dispersed, and the peaks were flat. The kurtosis values of the diameter distribution at breast height and height of the SB study area trees were less than zero, indicating that the distribution of diameter at breast height and crown width was more dispersed, and the peaks were flat. The distribution of woody plants in the BJ study area was more concentrated, while the SB study area was more scattered. In addition, the HJ study area had the most significant tree height distribution with kurtosis and skewness values greater than zero, indicating that its tree height distribution was to the left of the distribution curve and had a spike shape, saying that the tree height distribution of the HJ study area woody plants was more concentrated compared to the diameter at breast height and crown width.

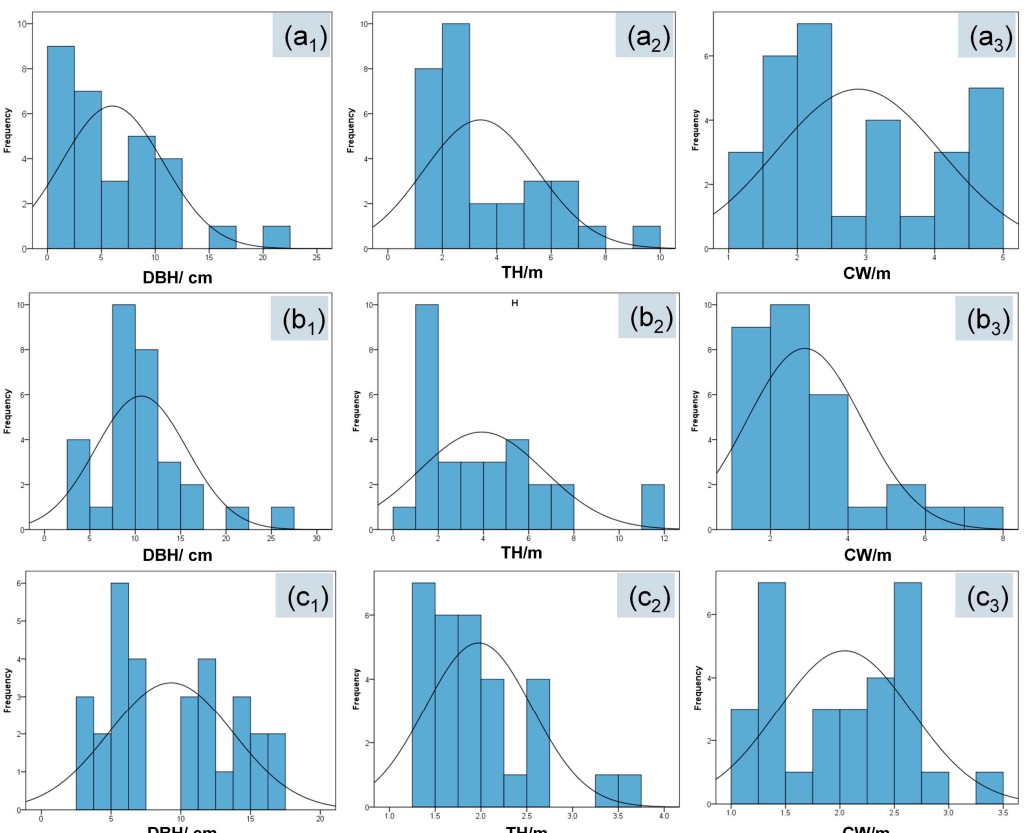

**Figure 8.** Nonspatial structure in MWLs. DBH, TH, and CW in the figure indicate the diameter at breast height (cm), tree height (m), and crown width (m), respectively (the same in the figure below); in addition, (**a₁**–**a₃**) indicate the BJ, (**b₁**–**b₃**) indicate the HJ, (**c₁**–**c₃**) indicate the SB.

### 3.2.2. Spatial Structure

In this section, we describe the spatial structural characteristics of three agroforestry systems (AFS) types. They are homegarden (HG), agrisilviculture (ASV), and multipurpose woodlots (MWLs).

Spatial Structural Characteristics of the HG

In the HG (Figure 9), first, the distribution pattern of stand level shows that the average angular scale distribution frequency in the BJ study area was slowly decreasing between $W = 0.25$ and $W = 1$. The distribution frequency (33.33%) was maximum and equal at $W = 0.25$ and $W = 0.5$, indicating that woody plants were primarily in a random and uniform distribution. The frequency of distribution (33.33%) was the most extensive and equal in the HJ study area at $W = 0.25$ and $W = 0.75$, respectively, and the frequency of distribution (26.67%) was relatively small at $W = 0.5$, indicating that the spatial distribution pattern of HJ mainly was in a random and uneven distribution (6.67%), the spatial distribution pattern of the SB study area was random and irregular.

Second, from the degree of species segregation, it can be seen that the distribution frequency was the largest at $M = 0.75$ (26.67%). The smallest at $M = 0.5$ (13.33%) in the BJ study area, and the proportion of the remaining mixed degree was 20%, indicating that the overall species segregation was strong, and the stand was relatively stable; the distribution frequency was the largest at $M = 0.5$ (33.33%) and $M = 0.25$ (26.67%) in the HJ study area. In addition, the frequency of distribution at $M = 0.75$ and $M = 0.1$ (20%) was equal, indicating that the study area has strong species segregation and a relatively stable stand; in the SB study area, the frequency of distribution at $M = 0.75$ was the highest (66.67%), and the frequency of distribution at $M = 0.5$ was the second highest (26.67%), indicating that the study area had a low species segregation and low stand stability. The distribution frequency

at M = 0.5 was the second highest (26.67%), implying that all tree species in the study area were quite isolated and it had low stability.

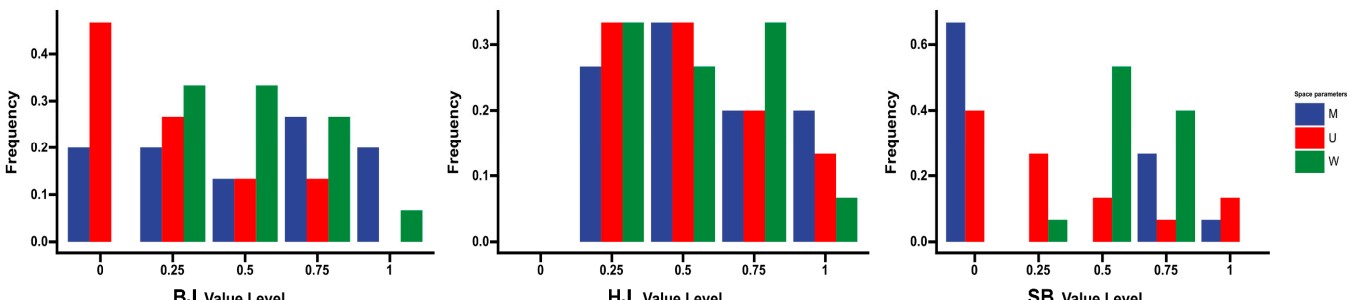

**Figure 9.** Spatial structure of HG. The W stands for angular scale, M stands for mixing degree, and U stands for the size ratio.

Finally, regarding the degree of stand size differentiation, the proportion of single trees with absolute dominance in the BJ study area reached 46.67%, while those at a complete disadvantage were 0. The average size ratio (0.52), which was between 0.33 and 0.67, indicated that the reference trees were mostly in an intermediate state; the HJ study area showed a trend of decreasing with the increase of the value level from absolute dominance to absolute disadvantage. The highest distribution frequency (33.33%) was found in the HJ study area at U = 0.25 and U = 0.5, indicating that the reference trees were mostly moderate; the proportion of single trees in absolute dominance in the SB study area reached 40% and showed a decreasing trend from dominance to complete sovereignty, indicating that the number of reference trees in this study area was gradually decreasing from dominant to absolutely inferior trees.

Spatial Structural Characteristics of the ASV

In the ASV (Figure 10), first of all, the distribution pattern at the stand level shows that the most significant distribution frequency (46.67%) was at W = 0.50 in the BJ study area. In comparison, the distribution frequency was the same at 26.67% at W = 0.25 and W = 0.75, and there was no W = 1.00, indicating that woody plants in this agroforestry type primarily reside in a random and uneven distribution; the distribution frequency in the HJ study area was the largest (46.67%) at W = 0.25. In the HJ study area, the distribution frequency was the highest at W = 0.25 (46.67%), and the distribution frequencies were equal at W = 0.5 and W = 0.75 (26.67%), indicating that woody plants in this type of agroforestry were mostly randomly and unevenly distributed; in the SB study area, the distribution frequency was the highest at W = 0.5 (66.67%), indicating that more tree species in this type of agroforestry were randomly and evenly distributed.

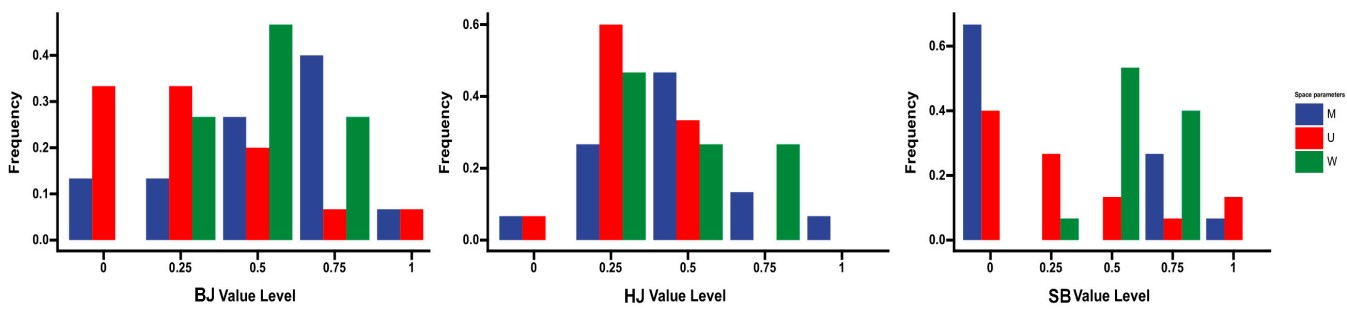

**Figure 10.** Spatial structure of ASV. The W stands for angular scale, M stands for mixing degree, and U stands for the size ratio.

Secondly, the distribution frequency at M = 0.75 was the highest in the BJ study area (40%). It shows that this type of agroforestry has a high degree of species isolation and relatively stable forest stands. The distribution frequency at M = 0.5 was the highest in the HJ study area (46.67%). The trend decreased from M = 0.5 to both ends, indicating a substantial degree of species segregation and a relatively stable stand in this type of agroforestry. The distribution frequency was the highest at M = 0 (33.33%) in the SB, indicating that the degree of species segregation in this type of agroforestry was low, and the stand was unstable.

Finally, regarding the degree of stand size differentiation, the proportion of woody plant size ratios in the BJ study area that were dominant and subdominant was the same (33.33%), indicating that the reference trees in this system were dominant. Subdominant species were the same, indicating that the number of reference trees in inferior and absolute inferior strains was low, and all were in the transition between dominant and subdominant to intermediate state; the proportion of single trees in complete dominance in the SB study area was 30%, while the proportion of single trees in subdominants was 30%. The balance of single trees at absolute authority reached 30%, while the proportion of single trees at subdominant and moderate was the lowest (33%). The ratio of single trees at inferiority and absolute inferiority (20%) indicates that the number of reference trees at inferiority and absolute inferiority was low. The overall state was excessive, from dominance to inferiority.

Spatial Structural Characteristics of the MWLs

In MWLs (Figure 11), first, the distribution pattern at stand level shows that the frequency of distribution was maximum (33.33%) at W = 0.50 in the BJ study area, equal at W = 0.25 and 0.50, and minimum (13.33%) at W = 1.00, indicating that woody plants in this study area were mostly in random and uneven distribution; the frequency of distribution was maximum (40%) at W = 0.75 and minimum (13.33%) at W = 1.00 in the HJ study area, indicating that woody plants in this agroforestry type were primarily in random and uneven distribution. The distribution frequency was the highest at W = 0.75 (40%) and the lowest at W = 1.00 (13.33%) in the HJ study area, indicating that woody plants in this agroforestry type were mostly randomly and unevenly distributed. It indicates that woody plants in AFS were mostly randomly and uniformly distributed and show a decreasing trend from a uniform distribution to random and uneven distribution.

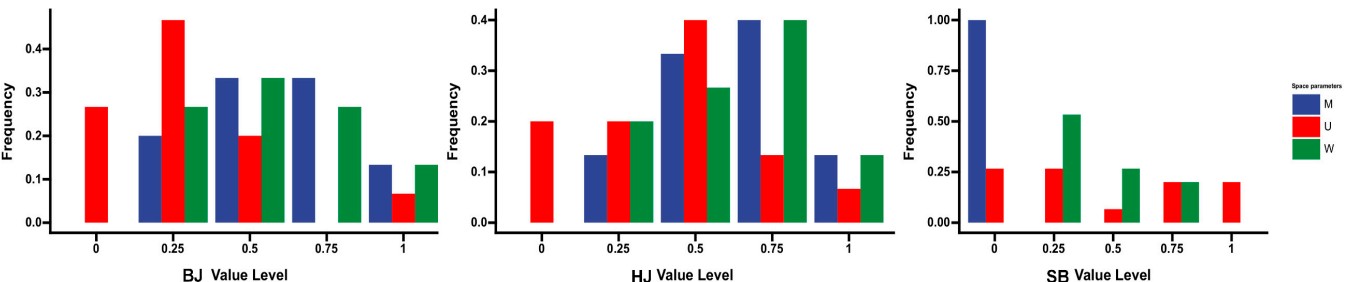

**Figure 11.** Spatial structure of MWLs. The W stands for angular scale, M stands for mixing degree, and U stands for the size ratio.

Second, in terms of species segregation, the distribution frequencies of M = 0.5 and M = 0.75 in the BJ study area were the highest and the same (33.33%), indicating that all types of AFS in the study area were in a relatively large proportion of medium-intensity mixing, indicating that the overall species segregation was strong, and the forest stands were relatively stable. The frequency of M = 0.5 and M = 0.75 in the HJ study area was the highest (33.33%), and there was a decreasing trend from the center of M = 0.75 to the two ends, indicating that the tree species in this type of agroforestry ecosystem were more segregated and the stand was more stable. The maximum distribution frequency (100%) was found at M = 0.00 in the multifunctional agroforestry ecosystem, indicating that species segregation was extremely low, and the stability was low in this study area.

Finally, in terms of stand size differentiation, the distribution frequency of U = 0.25 was the largest in the BJ study area (46.67%), followed by the distribution frequency at U = 0.00 (26.67%), which showed a "single-peaked" and decreasing trend distribution from the center to both ends of the subdominant distribution state. The proportion of absolute dominant and subdominant trees was the same in the SB study area (26.67%), the proportion of moderate trees was the lowest (6.67%), and the balance of inferior and absolute inferior trees was the same (20%). This indicates that the reference species in ASV were mostly dominant and subdominant trees.

### 3.3. Stability Analysis

3.3.1. Calculation of Evaluation Index Weights and Analysis of the Results

We can draw the following conclusions based on the weight values of each indicator in Table 2. First, soil fertility had the most significant contribution in the first-level hand with a weight of 0.3402, which is an essential factor affecting the AFS stability. In the first-level index of soil fertility, the weight of the soil's total potassium content was 0.0886, which is an extremely significant influencing factor. Potassium, phosphorus, and organic matter content in soil are essential nutrients in soil fertility, which play a decisive role in plant growth and development.

**Table 2.** Weights of indicators at each level.

| First-Class Index | | | Second-Class Index | | |
|---|---|---|---|---|---|
| Ordinal | Indicator Name | Weight | Ordinal | Indicator Name | Weight |
| | | | 1 | $U_{11}$ | 0.0462 |
| | | | 2 | $U_{12}$ | 0.0722 |
| | | | 3 | $U_{13}$ | 0.0374 |
| 1 | $U_1$ | 0.2983 | 4 | $U_{14}$ | 0.0274 |
| | | | 5 | $U_{15}$ | 0.0604 |
| | | | 6 | $U_{16}$ | 0.0546 |
| | | | 7 | $U_{21}$ | 0.0466 |
| | | | 8 | $U_{22}$ | 0.0323 |
| 2 | $U_2$ | 0.1909 | 9 | $U_{23}$ | 0.0366 |
| | | | 10 | $U_{24}$ | 0.0754 |
| | | | 11 | $U_{31}$ | 0.0547 |
| | | | 12 | $U_{32}$ | 0.0550 |
| | | | 13 | $U_{33}$ | 0.0567 |
| 3 | $U_3$ | 0.3402 | 14 | $U_{34}$ | 0.0886 |
| | | | 15 | $U_{35}$ | 0.0301 |
| | | | 16 | $U_{36}$ | 0.0300 |
| | | | 17 | $U_{37}$ | 0.0250 |
| | | | 18 | $U_{41}$ | 0.0587 |
| 4 | $U_4$ | 0.1706 | 19 | $U_{42}$ | 0.0658 |
| | | | 20 | $U_{43}$ | 0.0461 |

Second, the contribution of stand structure in the first-level index was second only to soil fertility, with a weight value of 0.2983, which is also a key factor affecting the AFS stability. Among the first-level indicators of stand structure, the degree of stand mixing with a weight of 0.0722 was an important influence factor in this indicator, which describes the degree of spatial segregation of tree species in AFS, interspecific relationships. The degree of stand mixing not only influences the nutrient uptake and photosynthesis of a single stand but also significantly impacts the growth and development of shrubs, herbs, and soil animals in the understory, which plays an essential role in the configuration of the agroforestry ecosystem.

Once again, the weight value of plant species diversity in the primary index was 0.1909, in which the diversity index and richness index were significant influencing factors. Finally, among the topographic factors, elevation was the main influencing factor; however,

the contribution rate of each secondary index is in a more balanced state, and they were all critical influencing factors in stability.

In conclusion, it is clear from the weight values of each indicator that there was not a single factor driving the stability of AFS, but all the influencing factors worked together. Therefore, in the planning and management of AFS, each indicator's contribution and influencing factors need to be considered comprehensively to achieve the stability and sustainable development of the system.

### 3.3.2. Multilevel Fuzzy Comprehensive Evaluation

Based on the evaluation level criteria, the evaluation affiliation matrix R of the evaluation index set Ui to the evaluation domain V was determined. Establishing the evaluation affiliation R is the key to fuzzy comprehensive evaluation. According to the effects of the evaluation fingers, the fuzzy relationship matrix affecting different AFS stability in the KD areas can be obtained by substituting the above affiliation function formula. According to the established fuzzy relationship matrix and the weight values of the secondary evaluation indexes (Table 2), the second evaluation was conducted using equation (24), and the secondary fuzzy evaluation matrix S11 of the first AFS in the BJ study area is obtained.

$$S_{11} = \begin{bmatrix} 0.0000 & 0.0000 & 0.2987 & 0.1603 & 0.0000 \\ 0.1895 & 0.3380 & 0.0000 & 0.0000 & 0.0000 \\ 0.0442 & 0.0662 & 0.0681 & 0.0382 & 0.0650 \\ 0.2857 & 0.0000 & 0.0677 & 0.0000 & 0.0128 \end{bmatrix}$$

Similarly, the secondary fuzzy evaluation matrix of the other two types of AFS can be obtained as:

$$S_{12} = \begin{bmatrix} 0.0211 & 0.1291 & 0.2381 & 0.0377 & 0.0000 \\ 0.0645 & 0.2623 & 0.1309 & 0.0000 & 0.0000 \\ 0.0442 & 0.1276 & 0.0000 & 0.1968 & 0.0000 \\ 0.0000 & 0.2497 & 0.0656 & 0.0000 & 0.0035 \end{bmatrix} S_{13} = \begin{bmatrix} 0.0000 & 0.0000 & 0.2003 & 0.2588 & 0.0000 \\ 0.0000 & 0.2399 & 0.2032 & 0.0961 & 0.0000 \\ 0.0804 & 0.0675 & 0.1276 & 0.0000 & 0.0407 \\ 0.3442 & 0.0624 & 0.0000 & 0.0000 & 0.0406 \end{bmatrix}$$

The secondary level fuzzy evaluation matrix of the AFS in HJ is obtained as:

$$S_{21} = \begin{bmatrix} 0.3227 & 0.1412 & 0.1221 & 0.0827 & 0.0000 \\ 0.2283 & 0.0995 & 0.0000 & 0.0765 & 0.0000 \\ 0.3107 & 0.1710 & 0.0000 & 0.0000 & 0.0290 \\ 0.2858 & 0.0000 & 0.1822 & 0.0000 & 0.1363 \end{bmatrix} S_{22} = \begin{bmatrix} 0.0000 & 0.0607 & 0.2646 & 0.0397 & 0.0000 \\ 0.0000 & 0.0000 & 0.0000 & 0.0000 & 0.3327 \\ 0.0163 & 0.1661 & 0.0715 & 0.0709 & 0.0120 \\ 0.3805 & 0.0209 & 0.0000 & 0.0000 & 0.1318 \end{bmatrix}$$

$$S_{23} = \begin{bmatrix} 0.0000 & 0.1533 & 0.1068 & 0.2686 & 0.0000 \\ 0.2277 & 0.1095 & 0.0000 & 0.0852 & 0.0000 \\ 0.0898 & 0.0219 & 0.0000 & 0.1543 & 0.0000 \\ 0.3455 & 0.0000 & 0.0000 & 0.0000 & 0.3585 \end{bmatrix}$$

The secondary level fuzzy evaluation matrix of the AFS in SB is:

$$S_{31} = \begin{bmatrix} 0.0000 & 0.0000 & 0.1660 & 0.1896 & 0.0784 \\ 0.0545 & 0.4597 & 0.0000 & 0.0000 & 0.0000 \\ 0.1108 & 0.0642 & 0.1809 & 0.0528 & 0.0000 \\ 0.0000 & 0.2608 & 0.1777 & 0.0000 & 0.0000 \end{bmatrix} S_{32} = \begin{bmatrix} 0.0000 & 0.0620 & 0.0690 & 0.0801 & 0.1003 \\ 0.0000 & 0.0000 & 0.1326 & 0.1917 & 0.0404 \\ 0.1073 & 0.0000 & 0.1212 & 0.0613 & 0.0934 \\ 0.0288 & 0.2469 & 0.1822 & 0.0000 & 0.0000 \end{bmatrix}$$

$$S_{33} = \begin{bmatrix} 0.0000 & 0.0000 & 0.1537 & 0.0439 & 0.0241 \\ 0.0000 & 0.0000 & 0.1230 & 0.2366 & 0.0000 \\ 0.0953 & 0.2046 & 0.1164 & 0.0785 & 0.0000 \\ 0.0000 & 0.2454 & 0.0000 & 0.1327 & 0.0000 \end{bmatrix}$$

### 3.3.3. Distribution of Stability Levels

When conducting the first-level evaluation, we used the Formula (25) based on the second-level fuzzy evaluation matrix S1 and the first-level evaluation index weight values

W (Table 2). The affiliation degree A value is calculated, the level is determined according to the principle of maximum affiliation, and the results are shown in (Table 3).

**Table 3.** Evaluation results.

| Structure Type | Area | Level | | | | | Rating |
|---|---|---|---|---|---|---|---|
| | | I | II | III | IV | V | |
| HG | BJ | 0.1000 | 0.0870 | 0.1238 | 0.0608 | 0.0243 | III |
| | HJ | 0.2929 | 0.1193 | 0.0675 | 0.0393 | 0.0331 | I |
| | SB | 0.0481 | 0.1541 | 0.1414 | 0.0745 | 0.0234 | II |
| ASV | BJ | 0.0336 | 0.1746 | 0.1072 | 0.0782 | 0.0006 | II |
| | HJ | 0.0705 | 0.0809 | 0.1032 | 0.0360 | 0.0901 | III |
| | SB | 0.0414 | 0.0606 | 0.1183 | 0.0813 | 0.0694 | III |
| MWLs | BJ | 0.1330 | 0.0741 | 0.0319 | 0.1489 | 0.0612 | IV |
| | HJ | 0.0324 | 0.1115 | 0.1089 | 0.1076 | 0.0072 | II |
| | SB | 0.0861 | 0.0794 | 0.1420 | 0.0772 | 0.0208 | III |

　　　First, in the HG stability evaluation grade, the affiliation degree (0.1238) of the BJ study area corresponding to the evaluation grade III was the largest and the level to which it belongs in general. The affiliation degree (0.2929) of the HJ study area corresponding to the evaluation grade I was the largest, and the level to which it belongs is excellent. The SB study area's affiliation degree (0.1541) related to the evaluation grade II was the largest, and the level to which it belongs is good. Second, among the ASV stability evaluation levels, the BJ study area had the largest affiliation (0.1746) corresponding to evaluation level II and belongs to the level of good. The HJ study area had the most significant association (0.1023) related to evaluation level II and belongs to the reasonable level. The SB study area had the largest affiliation (0.1183) corresponding to evaluation level III and belongs to the level of the fair. Finally, among the stability evaluation levels of MWLs, the BJ study area had the highest affiliation (0.1489) for evaluation level IV, and the affiliation level is poor. The HJ study area had the highest affiliation (0.1115) for evaluation level II, and the affiliation level is good. The SB study area had the highest affiliation (0.1420) for evaluation level III, and the affiliation level is fair.

　　　In conclusion, there were significant differences in the evaluation grades of the AFS stability of the same type in different KD control demonstration areas. It indicates that the maintenance of the AFS stability and the intensity of anthropogenic disturbance may have some correlation.

## 4. Discussion

### 4.1. AFS Species Composition and Species Diversity

　　　We found that the species composition of the AFS of KD control areas is simple, and the diversity level is low according to the data analysis (For details, please refer to Tables S7–S9 of the Supplementary Materials), the results of this study better support our hypothesis (i). The species composition was mainly shown in the configuration with crops, livestock breeding, or pasture crops, with fewer species of woody plants, and their diversity levels were lower than those of understory species. Similar findings have been better explained in related research reports. For example, studies on the characteristics of karst biodiversity and its reconstruction mechanism pointed out that there are few plant species (1–4 species) in each community in karst areas, and the species diversity level is generally low. Especially in some plantation forest areas planted to KD control, the species diversity level of vegetation is deficient [52–54]. Combined with the actual conditions in KD areas, the following mechanisms can be supported to explain this result: the gradient stress hypothesis, due to the existence of different gradients and stresses in the KD areas, bearing the multiple stresses of topography, karst drought, and high soil calcium, changing their structure as well as physiological and biochemical processes in the process of survival, through a complex and diverse response mechanism (cross-adaptation) to gain resistance to

adapt to the environment [55,56]. Therefore, favorable interactions and complementarities among species face an extraordinary test.

### 4.2. AFS Structural Features

Through data analysis (Figures 6–11), we obtained the following main conclusions: first, in terms of non-spatial structure, the distribution curves of diameter at breast height, tree height, and crown width of woody plants in different types of AFS under KD control areas were generally left and monoclinic, but HG was usually better than the remaining two types. Second, in terms of spatial structure, the spatial distribution pattern of woody plants in the HG and MWLs was mostly random and uneven distribution, with better segregation of tree species and more stable stands. In contrast, the ASVs were mainly in a uniform distribution, with relatively low stand isolation and poor stability. The above conclusions better support our hypothesis (ii) and have been better explained in the existing studies. On the one hand, with the growth and development of the stand, the demand for sunlight, water, and nutrients increases, which is subject to self-thinning and some human interference, making the stand aggregation weaker and will eventually develop into a random distribution pattern [57]. On the other hand, habitat heterogeneity and seed dispersal limitations force tree populations to aggregate and thus increase their survival potential [58]. In addition, the species composition and their number influence the degree of isolation of tree species and their stability levels, which are mainly related to species composition and biological characteristics [59]. We also argue that in poor KD areas, farmers' perceptions of agroforestry affect their management strategies. For example, failure to achieve the expected economic income in the short-term triggers them to change their planting strategy by changing tree species or directly cutting them down, thus leading to a reduction in tree species composition and a return to a single agroforestry ecosystem, which will be detrimental to the development of agroforestry. Therefore, farmers' willingness to participate should be added to the management strategy, which is similar to the view of agroforestry development in Ethiopia [60].

### 4.3. Driving Factor of AFS Stability

The distribution of the weight values of the primary indicators (Table 2) reflects the AFS stability constraints rather than solely the vulnerability of a singular karst ecological environment. They mainly consist of the interaction of site conditions such as soil physical and chemical properties and topographic factors and AFS structural factors such as species diversity and stand structure. This conclusion is somewhat different from our hypothesis (iii). However, it is better to show that the scientific analysis method can identify the influence of factors of stability and thus effectively explore the improvement strategies for AFS stability.

As far as the site conditions are concerned, in the KD area, on the one hand, due to the mismatch in the spatial distribution of soil water and soil resources, high spatial and temporal heterogeneity of water and heat factors, and extreme deficiency of nitrogen, phosphorus, and potassium, and in the process of vegetation restoration, with the accumulation of biomass and changes in species, the reserves of nitrogen, phosphorus, and potassium in the soil will decrease, which is hugely limiting for plant self-growth and diversity is highly limiting. In addition, the strongly developed karst dual hydrogeological structure leads to the unique "karst drought" phenomenon [61], the mosaic of rocks and shallow soils, the high heterogeneity of karst habitats and the significant differences in soil ecological functions, and the tendency of the microenvironment in KD areas to dry out and heat [62], lead to the development of AF in KD areas and a low survival rate and species diversity of AFS in KD areas [63]. Therefore, although studies have shown that AFS play a crucial role in providing ecosystem services [11,64], they are subject to significant stress effects on plant growth due to KD site conditions, resulting in low AFS stability.

As far as AFS structure is concerned, the higher number of plants and the greater the mixedness index, the higher the degree of isolation of its species and the more stable

the forest stand; the influence of plant species diversity on ecosystem stability should not be ignored. Ecological theory suggests that ecosystem stability is closely related to plant diversity [65,66]. The higher the number of species in an ecosystem, the higher its level of diversity, the more complex its trophic structure will be, the more resistant it will be to external disturbances, and the more stable it will be [9,67,68]. Therefore, plant stands construction and plant species diversity also influence stability factors.

### 4.4. Stability Improvement Strategy

According to the stability evaluation results, we can see that (Table 3) there is variability in the stability classes of different types of the AFS, among which, in HG, it mainly shows the ranking HJ (excellent) > SB (good) > BJ (fair); for ASV, it shows the ranking BJ (good) > HJ (fair) = SB (fair); for MWLs, it shows the ranking HJ (good) > SB (fair) > BJ (poor). Overall, this reflects HG > ASV > MWLs. Based on the above findings, combined with previous practical experience in AFS structure optimization and stability improvement, more targeted strategies are discussed around the relationship between species composition with its diversity and stability and the current situation of the AFS in KD control for reference stability enhancement strategies.

### 4.4.1. Species Composition and Its Diversity Strategy

Regarding species diversity and complexity, the core of biodiversity is species diversity, which enables ecosystems with higher diversity to use limited resources more fully and maintain their productivity at a higher level through complementary ecological niches [69]. As a result, different species have complex linkages in their quantitative and spatial, and temporal distributions, and species' trait differences, their respective survival strategies, and the relationships and interactions among species have essential effects on population survival and development dynamics and community structure, function, and succession [70–72]. In addition, the stability and diversity hypothesis theory suggests that desert vegetation communities are less stable due to the lack of redundant species, and their ecological functions are mainly carried out by dominant or community-building species, with a low number of other species [73,74]. Because of this, vegetation community simplicity, diversity, and structural indicators are low in AFS stability in KD. Therefore, it is essential to increase the number of plant community species in maintaining stability and pay special attention to introducing some fast-growing community dominant or community-building species to configure so that AFS generates disturbance resistance and resilience stability to maintain the strength of its system itself.

### Species Composition Strategy in the HG

The composition of woody plants and crops in HG primarily resides in the random selection of farmers and rarely follows a specific rule or guiding scheme for configuration. Regarding site conditions, they are mainly distributed around the courtyard with certain practicality and ornamental properties. It is primarily based on physiological and ecological growth habits of long-term competition, adaptation, and evolution of a mosaic pattern of different woody plants, crops, and other vegetation. Its dominant species are not prominent, primarily common species, and most of the plants are calcium-loving, lithophytic, drought-tolerant, and barren [15]. Due to the complexity of KD habitats, there are not only one species that can adapt to the same successional stage habitat but also a species group composed of multiple species with the same adaptations [38,75]. Therefore, if the dominance of the dominant tree species does not decrease, different colors and seasonal phases can be changed according to the season and phenology of the tree species. The temporal relationship structure can be fully utilized to appropriately increase the native ornamental tree species to improve their species diversity level and thus enhance the beauty of the landscape.

Species Composition Strategy in the ASV

The ASV is mainly located in or around farmland to maintain farmers' economic income while taking ecological benefits into account. The species composition should consider the types of products and their uses and select native species to meet the needs of local people and markets. However, it should be regarded that woody plants and herbaceous crops can improve the soil and the small environment, which is beneficial for enhancing the fertility of the agroforestry land and thus increasing species diversity. The KD areas are locally fragile. The AF in KD areas should follow the unique local natural environment and moisture conditions and select complementary, less water-demanding drought-tolerant plants. In addition, the thin soil layer and poor continuity in KD areas should select for trees that can grow in rock crevices and ravines [76–78].

Species Composition Strategy in the MWLs

The MWLs are mainly balanced to maintain ecological, economic, and social functions and contain some of the main tasks of HG and ASV, such as timber production and food production. For this type of AFS, crop species, and woody plants should be reasonably selected and deployed, and various woody plants can be configured together. Still, the inhibitory effect between plants should be considered. In terms of spatial utilization, we should choose to match the height of trees, shrubs, and grasses and to match loose and compact plants; in terms of root depth, we should choose to match deep-rooted and shallow-rooted crops, which can efficiently utilize water and nutrients at different levels in the soil and choose shade-tolerant crops with better shade woody plants to fully use light energy resources for light energy utilization. The AFS comes after intercropping, crop set, or strip planting in the collocation [16,20]. Implementing intercropping or strip planting will change the group structure of crops, create edge row advantage, and allow good ventilation and light conditions for plants.

4.4.2. Artificial Interference Strategy

Regarding anthropogenic disturbances, KD areas are locally fragile. In its ecological context of exposed bedrock, shallow soil, and marked water infiltration, it must undergo strict natural screening to enable plant populations with calcium-loving, drought-tolerant, and lithophytic characteristics to survive [78]. Therefore, the species selection and configuration should be based on various highly resistant woody plants, giving full play to their resistance to disturbance stability in the KD environment against spatial heterogeneity of small habitats, drought stress, high calcium, and pests and diseases. It should carry out the reasonable transformation of multi-species, appropriate trees for the right place, take native species as auxiliary species and appropriately increase the numbers of species of trees and shrubs so that the structure and function of each species are complementary. Build a rich diversity of AFS. In addition, configure decoy tree species and set up isolation zones; implement pruning, nurturing, truncation, and sanitary felling. The theory of creating AFS with multi-species configuration and the technique of implementing dynamic management of agroforestry ecosystem is developed. Strengthen the role of natural enemies and simultaneously carry out appropriate chemical and biological control measures. On the one hand, we should try to avoid the occurrence of harmful human interference and reduce the factors that cause AFS instability; on the other hand, we should strengthen the management of stand nurturing and ecosystem management, enhance the screening and trait improvement of highly resistant strains of trees, and fundamentally change the low-quality afforestation.

**5. Conclusions**

Our research has quantified the AFS structural characteristics by descriptive statistics and structural parameter calculation methods. The AFS stability in the KD evaluation index system was constructed using the fuzzy comprehensive evaluation method, the weights of each index were calculated by the entropy weight method, and the stability was evaluated and graded by establishing an evaluation set. The structure and stability influencing factors

were explored. The structure optimization and stability enhancement strategies were proposed to provide a reference basis for studying AFS's sustainable operation and service supply capability. The main conclusions are as follows:

(i) The species composition of the AFS in the KD control areas has a simple structure, the overall diversity level is low, and the diversity level of herbaceous plants is better than that of woody plants.

(ii) The overall distribution curves of diameter at breast height (DBH), tree height (TH), and crown width (CW) of woody plants in the AFS in the KD control areas were slight to the left, with a single-peaked pattern, mostly randomly and unevenly distributed in space, with a low degree of tree species isolation and relatively weak stand stability.

(iii) There is variability in the stability classes of different types of AFS, among which, in HG, it mainly shows the ranking in study areas of HJ (excellent) > SB (good) > BJ (fair); in ASV, it shows BJ (good) > HJ (fair) = SB (fair); in MWLs, it shows HJ (good) > SB (fair) > BJ (poor). Overall reflecting HG > ASV > MWLs.

(iv) When structural optimization was applied, corresponding measures can be taken according to farmers' wishes for different AFS and their primary business purposes. The improvement of stability depends mainly on the utility of structural optimization coupled with positive human interference (for example, pruning, dwarfing, and dense planting). This study provides a scientific reference for maintaining the stability and sustainable development of the AFS in the KD control areas.

Our study still needs to be improved in that the drivers of the structure and stability of AFS in KD control areas are not only biological and environmental interactions. Although anthropogenic disturbances were mentioned in our manuscript, no specific analysis was made. Further explanation of the effects of anthropogenic disturbances on KD-managed AFS is necessary for future studies.

**Supplementary Materials:** The following supporting information can be downloaded at https://www.mdpi.com/article/10.3390/f14040845/s1. Table S1: Geographical information of AFS sample sites; Table S2: Principles and description of the construction of the index system; Table S3: Stability evaluation index screening criteria and meaning; Table S4: AFS Stability evaluation indicators and their ecological significance; Table S5: Evaluation Indicators; Table S6: Evaluation index standard grading; Table S7: AFS plant composition and importance values; Table S8: Changes in ecosystem plant composition; Table S9: AFS species diversity index.

**Author Contributions:** Conceptualization, S.J.; methodology, S.J. and J.X.; software, S.J., and J.X.; validation, S.J.; formal analysis, S.J.; and J.X.; investigation, S.J., Y.Y., Y.H.; Z.W. and J.X.; resources, S.J.; data curation, S.J.; writing—original draft preparation, S.J.; writing—review and editing, S.J. and J.X.; visualization, S.J.; supervision, K.X.; project administration, K.X.; funding acquisition, K.X. All authors have read and agreed to the published version of the manuscript.

**Funding:** This study was supported by the Key Project of Science and Technology Program of Guizhou Province (No. 5411 2017 Qiankehe Pingtai Rencai), the China Overseas Expertise Introduction Program for Discipline Innovation (No. D17016), and the National Major Research and Development Program of China (2016YFC0502607).

**Institutional Review Board Statement:** Not applicable.

**Informed Consent Statement:** Not applicable.

**Data Availability Statement:** Our data are placed in the Supplementary Materials.

**Acknowledgments:** Thanks to Shuzhen Song and Xiaobi Wu for our guidance in the experiment.

**Conflicts of Interest:** The authors declare no conflict of interest.

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
