# Peer review of "Agroforestry Ecosystem Structure and the Stability Improvement Strategy in Control of Karst Desertification"

_forests, doi:10.3390/f14040845_

Round 1

Reviewer 1 Report

In my opinion, the manuscript is very interesting for the scientific community of international readers. The authors have written it nicely and provided valuable information in this manuscript.  All my comments are below:

All my comments are below:

Comments:

Abstract

The abstract of the manuscript must be rewritten with an emphasis on the major finding of the study as one paragraph. 

Introduction

The introduction of the manuscript is written very nicely by the authors. Please provide some information about these three studied AFS in the introduction section of the manuscript.

Materials and methods

The materials and methods section of the manuscript is also written very nicely, but some minor corrections are needed.

In line no 128: Please write once the full name of the word abbreviated as HG in the manuscript.

In line no 131: Please write once the full name of the word abbreviated as AS in the manuscript. 

In line no 132: Please write once the full name of the word abbreviated as MW in the manuscript.

In lines, no 146-149: Please write the Binomial name of the crops and fruit tree in the manuscript for the suitability of international readers. 

In lines, no 160-162: Please write the Binomial name of the crops and fruit tree in the manuscript for the suitability of international readers.

In lines, no 170-172: Please write the Binomial name of the crops and fruit tree in the manuscript for the suitability of international readers.

In lines, no 183-184: Please provide some information about the instruments used and methods adopted for measuring the tree diameter crown width and the height of the trees, shrub species, and herbaceous species.

In lines, no 196-199: Two soil samples were collected from the 0 to 10 cm depth separately at one time or two different times describe clearly. 

Please incorporate the photographs of the different studied AFS and their soil rocks.

In lines, no 197-198: Please explain the chemical analysis that the authors have done in the study.

In table 4 authors write the bulk density, but not mentioned how they have estimated the bulk density. Please mention the method adopted for the estimation of bulk density.

In table 4 authors writing, the term SOM, TN,  TP, and TK, but do not write the full name of these terms in the manuscript. Please must write the full and their abbreviation once in the manuscript.

Results

Table 7  Authors did not mention the type of plants i.e. Trees, shrubs, and herbs. Please incorporate it in tables.  

Table 10 statistical analysis has been done, but in materials and methods, nothing writes about this.  Please write about the statistical analysis which has been done by the authors.

What are the units of measurement of D, H, and W values given in table 10? Please add to the table.

In line numbers, 451 to 464, which results have been written and values were taken from which table? Please mentioned the table.

The values are given in table 10 and why make figure number 2, 3, and 4 these values again? Please delete the values from table 10 in my opinion. 

Please mentioned figure 5 in the result line numbers 524 to 614 in the proper place.

Line no 624, 628: The authors cite different references in the results section.  In my opinion, the references should be cited in the discussion section.

In my opinion, the results section may be shortened to properly understand the results.

Discussion

The authors cite tables 16 and 17 in the discussion, but the table is not given or it is a typographic mistake?

The discussion also needs to improve by adding proper results of the study and some new references related to the study. 

In the manuscript, crops are not included. In my view, agricultural crops should be included in the manuscripts.

Conclusion

The conclusion of the study may be shortened.

Author Response

Thanks for your valuable comments on our manuscript. We have revised our manuscript based on your comments, please to review it.

Reviewer 2 Report

Dear Authors.

The submitted manuscript titled „Agroforestry ecosystem structure and the stability improvement strategy in the karst desertification control” contains very interesting results. Neverthless, I have found some imperfections-which in my opinion- should be imroved or clarified.

General remark: text is difficult to follow due to frequent use of acronyms.

1.       In my opinion the Abstact section should be shorten. It should referr to main parts of manuscripts: aims, methods, results and conclusions.

2.       In chapter Introduction all abreviation should be explicated.

3.       Line 179. I encourage Authors to prepare the Figure with design of field trial. It would be very useful and helpful in understanding of field studies and data collection.

4.       Are data from BJ, HJ and SB compared statistically? If so, please add the subchapter statistical analysis.

5.       Presentation of results is very extent. I suggest to shorten it. Perhaps some Tables migt be moped into supplementary material.

6.       Figures 2-5 arre illegible, there is lack of axa description. The data indicated in Figures should be not duplicated in text.

Author Response

Thanks for your valuable comments on our manuscript. We have revised our manuscript based on your comments, and please to review it.

Round 2

Reviewer 1 Report

The authors, I am grateful for your hard work and did a great job reviewing the manuscript, thus improving its overall quality. Most of the comments are now incorporated and addressed by the authors in the manuscript. 

Author Response

Thank you again for your approval of our manuscript.

Reviewer 2 Report

Dear Authors,

In my opinion Your manuscript might be published in present form.

Author Response

(The authors gave the same response as above.)
